# Model free adaptive control of nonlinear second-order multi-agent systems based on backstepping under mixed attacks

1nd Dong Liu
*College of Automation*
*Shenyang Aerospace University*
Shenyang, China
sy04848@126.com

2st Lei Han
*College of Automation*
*Shenyang Aerospace University*
Shenyang, China
h20009262024@163.com

*Abstract*—This paper employs model-free adaptive control (MFAC) methods to investigate the trajectory tracking control problem of second-order nonlinear multi-agent systems (MASs) under mixed attacks. Initially, since mixed network attacks significantly impact system stability, predictive-based compensation mechanisms are designed to reduce the effects of these attacks. Furthermore, based on the backstepping method and MFAC techniques, virtual desired velocities are constructed to decompose the second-order multi-agent systems into two interconnected subsystems. Moreover, leveraging the desired states of each subsystem, distributed MFAC schemes are proposed using the backstepping approach to achieve trajectory tracking control of the second-order multi-agent systems. Finally, the effectiveness of the proposed method is validated through two simulation examples, demonstrating robustness and efficiency in countering the adverse impacts of mixed attacks.

*Index Terms*—mixed attack, data-driven control, second-order multi-agent systems, backstepping method

## I. Introduction

In recent years, multi-agent systems have seen widespread application and have emerged as a significant research direction in the field of network intelligence. These systems have achieved numerous outstanding research results in various practical fields, such as remote unmanned aerial vehicles (UAVs) [1], multirotor drone systems [2], and multi-robot systems [3]. Compared to traditional control systems, multi-agent systems offer several advantages, including distribution, autonomy, efficiency, and cost-effectiveness. These advantages make multi-agent systems particularly appealing for a wide range of applications, from industrial automation to environmental monitoring, and from defense systems to intelligent transportation networks.

Typically, current research on cooperative control problems can be classified into three primary directions: leaderless cooperative control [4], leader-follower cooperative control [5], and containment control [6]. Leaderless cooperative control focuses on achieving consensus among agents without a designated leader, emphasizing decentralized decision-making and robustness to individual agent failures. Leader-follower cooperative control, on the other hand, involves a hierarchical structure where certain agents (leaders) guide the behavior of the rest (followers), facilitating coordinated actions and efficient task completion. Containment control aims at ensuring that certain agents (leaders) steer the group of agents (followers) to stay within a desired area or follow a specific trajectory, which is crucial in applications such as formation flying and swarm robotics.

In [7], predictive control algorithms are applied to multi-agent systems with communication constraints. These predictive control algorithms enable multi-agent systems to achieve cooperative control even in the presence of network attacks. By anticipating future states and adjusting control inputs accordingly, predictive control enhances the system's resilience to disruptions and uncertainties. This is particularly important in scenarios where reliable communication cannot always be guaranteed, and the system must adapt dynamically to changing conditions and potential threats. The integration of predictive control algorithms thus represents a significant advancement in the capability of multi-agent systems to operate effectively in complex and adversarial environments, ensuring robust performance and maintaining desired operational goals despite the presence of network attacks.

It is worth noting that cybersecurity is critically important in multi-agent systems. Generally speaking, network attacks include denial of service (DoS) attacks, false data injection (FDI) attacks, deception attacks, and others. These network attacks often have various unexpected impacts on the system, disrupting normal operations and potentially causing significant damage. Therefore, in recent years, there has been increasing attention to the study of denial of service attacks. In [8], a cooperative security control strategy was proposed to allow for non-periodic occurrences of attacks. In [9], an event-triggered state observer was designed to treat attack signals as states. In [10], compensation for attacks was achieved by inversely compensating for deception attacks, mitigating their adverse effects on the system.

It is noteworthy that most existing studies on multi-agent systems under network attacks assume a fundamental condition that the system model is known. However, in many

complex systems, acquiring an accurate model is challenging, which significantly limits the depth and scope of research on cooperative control. Nonetheless, since Hou et al. [11] introduced the model-free control method based on input-output (I/O) data, further developed in [12]-[13], this issue has been addressed. The model-free control method relies solely on the systems input and output data, without requiring a specific mathematical model. This approach has opened new possibilities for the study of many complex systems, allowing researchers to explore cooperative control issues in greater depth. To date, a substantial body of research has employed this method to further investigate cooperative control problems [14]-[15]. However, for more complex second-order systems, this method still struggles to provide an adequate and comprehensive solution.

Backstepping is a pivotal control algorithm in the realm of nonlinear control. The specific design process, as described in reference [16], involves decomposing the model of the controlled object into multiple subsystems to ensure stability. Then, iterative design is applied to the intermediate virtual control quantities, followed by tracking the expectations of these virtual quantities to achieve desired control objectives. Reference [17] introduces a backstepping tracking control strategy based on state observers to enhance system performance. Reference [18] utilizes a backstepping approach with command control to avoid the complexity of intermediate virtual quantities, simplifying the control process. Reference [19] proposes a backstepping control strategy that addresses the computational complexity inherent in traditional backstepping control algorithms. Reference [20] combines backstepping with model-free control to solve control problems by decomposing second-order systems into two interconnected subsystems. In summary, the combination of backstepping and model-free control effectively addresses tracking control problems under external disturbances, providing robust solutions in complex environments.

Inspired by the aforementioned research results, this article explores leader-following control of second-order nonlinear multi-agent systems based on backstepping. It considers mixed network attacks and attack compensation mechanisms. In comparison with other articles in the field of multi-agent systems, the highlights of this article are:

(1) Proposing model-free adaptive control for second-order MASs using the backstepping method, this approach tracks and designs virtual desired velocities, decomposing the second-order nonlinear system into two interconnected serial subsystems, each designed without relying on models.

(2) Introducing lyapunov functions demonstrates that under the MFAC algorithm, actual velocities promptly track desired virtual velocities, achieving asymptotic stability of velocity tracking errors.

(3) Considering mixed network attacks in sensor-controller communication channels, including denial of service (DoS) attacks, false data injection (FDI) attacks, and deception attacks (DA), this study designs predictive-based attack compensation mechanisms to ensure system stability.

The main content of this paper will be elaborated in detail in the following sections. Section II introduces graph theory and problem formalization. Section III presents the controller design for second-order systems considering mixed network attacks and convergence analysis. Section IV provides a numerical case to demonstrate the control effectiveness of the proposed control scheme. Section V concludes the paper.

## II. GRAPH THEORY AND PROBLEM FORMULATION

### A. Graph Theory

The set of real numbers is denoted by $\mathbb{R}$. For any matrix $A \in \mathbb{R}^{N \times N}$, the norm is indicated as $\|A\|$. The notation $\mathrm{Diag}(\cdot)$ represents a diagonal matrix, and $I$ is the identity matrix with appropriate dimensions. Graph theory serves as a crucial framework in multi-agent systems to model interaction topologies. Consider a weighted directed graph $G = (V, E, A)$, where $V = \{1, 2, \ldots, N\}$ is the set of vertices, $E \subseteq V \times V$ is the set of edges, and $A$ is the adjacency matrix. The vertices $V$ also function as indices for the agents.

When agent $j$ receives a message from agent $i$, the edge $(i, j) \in E$ exists. Here, agent $j$ is referred to as the child of agent $i$, and agent $i$ is the parent of agent $j$. The neighborhood of agent $i$ is described as $\mathcal{N}_i = \{j \in V \mid (j, i) \in E\}$. The weighted adjacency matrix $A = (a_{i,j}) \in \mathbb{R}^{N \times N}$ is defined such that $a_{i,i} = 0$ and $a_{i,j} = 1$ if $(j, i) \in E$; otherwise, $a_{i,j} = 0$.

The Laplacian matrix $L$ of graph $G$ is defined as $L = D - A$, where $D = \mathrm{diag}(d_{in,1}, d_{in,2}, \ldots, d_{in,N})$. The in-degree $d_{in,i}$ of vertex $i$ is given by $\sum_{j=1}^{N} a_{i,j}$. A graph is termed strongly connected if a path exists between every pair of vertices.

### B. Problem Formulation

Considering the following second-order discrete-time nonlinear system:

$$
\begin{cases}
y_p(k+1) = h_p(y_p(k), x_p(k)) \\
x_p(k+1) = f_p(x_p(k), u_p(k))
\end{cases}
\tag{1}
$$

in this case, Additionally, $f_p(\cdot)$ and $h_p(\cdot)$ denotes an unknown nonlinear function.

*For System 1*

*Assumption 1* [21]: The system (1) adheres to the generalized Lipschitz condition, which implies that if $\Delta u_p(k) \neq 0$, then $\Delta x_p(k+1) \leq b_1 \Delta u_p(k+1)$, where $\Delta x_p(k+1) = x_p(k+1) - x_p(k)$ and $\Delta u_p(k+1) = u_p(k+1) - u_p(k)$. Here, $b_1$ is a positive constant.

*Assumption 2* [21]: The partial derivatives of $f_p(\cdot)$ with respect to the system control input $u_p(k)$ are continuous.

*Assumption 3*: The ratio of change in $u_q(k)$ to the change in $u_p(k)$ is bounded, i.e., $|\Delta u_q(k)/\Delta u_p(k)| < \varsigma$, where $\varsigma$ is a positive constant.

*Assumption 4* [22]: The directed graph $\mathcal{G}$ possesses a directed spanning tree.

*Remark 1:* Assumptions 1 and 2 are commonly found in control systems, facilitating the use of dynamic linearization methods in this article. Assumption 3 will be applied in

the subsequent proof. Assumption 4 is crucial for achieving leader-following consensus control.

*Lemma 1* [23]: For system (1), if Assumptions 1 and 2 are satisfied and $|\Delta u_p(k)| \neq 0$ for all time $k$, then a pseudo-partial derivative (PPD) parameter $\Phi_{p1}(k)$ exists. Consequently, system (1) can be expressed as:

$$\Delta x_p(k+1) = \Phi_{p1}(k)\Delta u_p(k) \qquad (2)$$

where $\Phi_{p1}(k)| \leq \bar{\mathfrak{b}}_1$

Similarly, for System 2 *Finally, for System 2*

*Assumption 5* [21]: System (2) adheres to the generalized Lipschitz condition, implying that if $\Delta x_p(k) \neq 0$, then $|\Delta y_p(k+1)| \leq b_2|\Delta x_p(k+1)|$, where $\Delta y_p(k+1) = y_p(k+1) - y_p(k)$ and $\Delta x_p(k+1) = x_p(k+1) - x_p(k)$. Here, $b_2$ is a positive constant.

*Assumption 6* [21]: The partial derivatives of $f_p(\cdot)$ with respect to the system control input $u_p(k)$ are continuous.

*Assumption 7*: The ratio of change in $x_q(k)$ to the change in $x_p(k)$ is bounded, i.e., $|\Delta x_q(k)/\Delta x_p(k)| < \zeta_1$ with $\zeta_1$ being a positive constant.

*Assumption 8* [22]: The digraph $\mathcal{G}$ possesses a directed spanning tree.

*Lemma 2* [23]: For system (2), if Assumptions 5 and 6 are satisfied and $|\Delta x_p(k)| \neq 0$ for all time $k$, then a pseudo-partial derivative (PPD) parameter $\Phi_{p2}(k)$ exists. Consequently, system (2) can be expressed as:

$$\Delta y_p(k+1) = \Phi_{p2}(k)\Delta x_p(k) \qquad (3)$$

where $\Phi_{p2}(k)| \leq \bar{\mathfrak{b}}_2$

For the overall system

$$\Delta y_p(k+1) = \Phi_p(k)\Delta u_p(k-1) \qquad (4)$$

where $\Phi_p(k) = \Phi_{p1}(k-1)\Phi_{p2}(k)$

The distributed output error for the $p$th agent is defined as follows:

$$\varepsilon_p(k+1) = \sum_{p \in \mathcal{N}_p} a_{pq}(y_p(k+1) - y_q(k+1)) \qquad (5)$$

By substituting equation (1) into equation (5) and introducing a new nonlinear function $\mathcal{F}(\cdot)$, equation (5) becomes:

$$\varepsilon_p(k+1) = \mathcal{F}_p(y_p(k), u_p(k), y_q(k), u_q(k)) \qquad (6)$$

*Theorem 1:* For equation (6), if assumptions 1-3 hold true and $|\Delta u_p(k)| \neq 0$ at all times $k$, then there exists a pseudo-partial derivative (PPD) parameter $\Phi_p(k)$. Thus, equation (6) can be rewritten as:

$$\Delta \varepsilon_p(k+1) = \Phi_p(k)\Delta u_p(k) \qquad (7)$$

where $\Delta \varepsilon_p(k+1) = \varepsilon_p(k+1) - \varepsilon_p(k)$ and $|\Phi_p(k)| \leq \mathfrak{b}_p$, $\mathfrak{b}_p$ is a positive constant.

Proof: Based on system (6), $\Delta \varepsilon_p(k+1)$ is calculated as

$$\begin{aligned}
\Delta \varepsilon_p(k+1) = {} & \mathcal{F}_p[y_p(k), u_p(k), y_q(k), u_q(k)] \\
& - \mathcal{F}_p[y_p(k-1), u_p(k-1), y_q(k-1), u_q(k-1)] \\
& + \mathcal{F}_p[y_p(k-1), u_p(k-1), y_q(k-1), u_q(k-1)] \\
& - \mathcal{F}_p[y_p(k-1), u_p(k-1), y_q(k), u_q(k)]
\end{aligned}$$

Then, using the differential mean value theorem for $\mathcal{F}_p[y_p(k), u_p(k), y_q(k), u_q(k)] - \mathcal{F}_p[y_p(k-1), u_p(k-1), y_q(k), u_q(k)]$ with respect to $u_p(k)$, one can get

$$\Delta \varepsilon_p(k+1) = \frac{\partial \mathcal{F}_p^*}{\partial u_p(k)}\Delta u_p(k) + \Psi_p(k)$$

where $\left(\frac{\partial \mathcal{F}_p^*}{\partial u_p(k)}\right)$ denotes the partial derivative value of $\mathcal{F}_p$ with respect to $u_p(k)$ in the interval $[u_p(k-1), u_p(k)]$. The term $\Psi_p(k)$ is defined as $\mathcal{F}_p[y_p(k-1), u_p(k-1), y_q(k), u_q(k)] - \mathcal{F}_p[y_p(k-1), y_q(k), u_q(k), u_p(k-1)]$.

Consider the data equation $\Psi_p(k) = \eta_p(k)\Delta u_p(k) + \eta_q(k)\Delta u_q(k)$ with variable $\eta_p(k)$ and $\eta_q(k)$ for each fixed time $k$. Based on Assumption 3, there exists a solution $\eta_p^*(k)$ such that $\Psi_p(k) = \eta_p^*(k)\Delta u_p(k)$ holds. Letting $\Phi_p(k) = \left(\frac{\partial \mathcal{F}_p^*}{\partial u_p(k)}\right) + \eta_p^*(k)$, the formula above can be derived. Furthermore, according to Assumptions 1 and 4, there exist constants $\bar{c}_p > 0$ and $\bar{c}_q > 0$ such that:

$$\begin{aligned}
|\Delta \varepsilon_p(k+1)| \leq {} & \bar{c}_p a_{pp}|\Delta u_p(k)| + \sum_{q \in \mathcal{N}_p} \bar{b}_q a_{pq}|\Delta u_q(k)| \\
\leq {} & \bar{c}_p a_{pp}|\Delta u_p(k)| + \sum_{q \in \mathcal{N}_p} \bar{c}_q a_{pq}|\Delta u_q(k)| \\
\leq {} & \left(\bar{c}_p a_{pp} + \sum_{q \in \mathcal{N}_p} \bar{b}_q a_{pq}\right)|\Delta u_p(k)| \\
= {} & c_p|\Delta u_p(k)|
\end{aligned}$$

where $c_p = \bar{c}_p a_{pp} + \sum_{q \in \mathcal{N}_p} \bar{b}_q a_{pq}|e|$. Thus, one can follow from the formula above that the PPD parameter $\Phi_p(k)$ is bounded, i.e., $|\Phi_p(k)| \leq c_i$.

## III. MAIN RESULTS

This section elaborates on the main content of this article, including the design of mixed network attacks, compensation mechanisms, controller design, and proof of bounded distributed output errors. The conceptual framework of the article is illustrated in Fig.1.

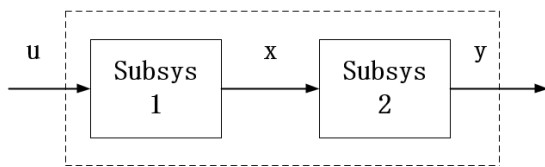

Fig. 1. System model decomposition diagram.

## A. mixed Cyber-attacks Design

Here, the design of a hybrid network attack will be elaborated. The attack, consisting of DoS, FDI, and DA, constitutes a stochastic attack. $\varepsilon_{pi}(k)$ represents the output signals under different attacks, with $\bar{\varepsilon}_p(k)$ denoting the final output signal subjected to the mixed attack.

When $i = 1$, the system suffers a Dos attacks, formula (5) can be written as:

$$\varepsilon_{p1}(k) = l_1(k)\varepsilon_p(k) \tag{8}$$

where $l_1(k)$ indicates the success of the DoS attacks and follows a Bernoulli distribution. If $l_1(k) = 0$, it indicates that the attacks were successful, with a probability of $\mathbb{P}\{l_1(k) = 0\} = \bar{l}$. Conversely, if $l_1(k) = 1$, the attacks failed, with a probability of $\mathbb{P}\{l_1(k) = 1\} = 1 - \bar{l}$.

When $i = 2$, the system experiences FDI attacks, and thus formula (5) can be rewritten as:

$$\varepsilon_{p2}(k) = \varepsilon_p(k) + \pi_p(k) \tag{9}$$

where $\pi_p(k)$ denotes the gain parameter of the FDI attacks, which varies randomly within a certain range. FDI attacks use $r_p(k) = \exp\{-\|\varepsilon_p(k) - \hat{\varepsilon}_p(k)\|\}$ to determine whether an attack has occurred. If $r_p(k)$ is less than a specific positive constant $\nu$, the attack is deemed to occur, resulting in $r_p(k) = 0$.

When $i$, the system is subjected to DA attacks, and formula (5) can be rewritten as:

$$\varepsilon_{p3}(k) = (-1)^{1-l_3(k)}\varepsilon_p(k) \tag{10}$$

where $l_3(k)$ indicates whether the DA attacks were successful, following a Bernoulli distribution. When $l_3(k) = 0$, it signifies that the attacks were successful, with a probability of $\mathbb{P}\{l_3(k) = 0\} = \bar{l}$. Conversely, when $l_3(k) = 1$, the attacks failed, with a probability of $\mathbb{P}\{l_3(k) = 1\} = 1 - \bar{l}$.

Ultimately, $l(k)$ indicates the success of a mixed attack, with $l(k) = 1$ representing a successful attack and $l(k) = 0$ indicating a failed attack. Here, $l(k) = l_1(k)l_2(k)l_3(k)$.

In summary, the following distributed output error formula with prediction compensation is proposed:

$$\bar{\varepsilon}_p(k) = l(k)\varepsilon_p(k) + (1 - l(k))\hat{\varepsilon}_p(k) \tag{11}$$

where $\hat{\varepsilon}_p(k) = (\bar{\varepsilon}_p(k-1) + \hat{\Phi}_{p2}(k-1)\Delta u_p(k-1))$

*Remark 2:* In this context, DoS attacks can be viewed as intentional disruptions targeting network protocol implementation, with the objective of making the computer or network unable to provide normal services or access resources. FDI and DA attacks can be considered as attackers injecting false signals to substitute genuine information, thus preventing the system from reaching its intended objectives.

## B. Design of Second-order System

For System 1, due to the compression of the nonlinear dynamic characteristics of the system into $\Phi_{p1}(k)$, obtaining its dynamic model remains challenging, but numerical variations can be estimated. Therefore, the cost function for $\Phi_{p1}(k)$ is given in the following form:

$$J(\hat{\Phi}_{p1}(k)) = |\Delta x_p(k+1) - \hat{\Phi}_{p1}(k)\Delta u_p(k-1)|^2 + \mu|\hat{\Phi}_{p1}(k) - \hat{\Phi}_{p1}(k-1)|^2$$

Taking the partial derivative of $\hat{\Phi}_{p1}(k)$ from the above equation yields 0, with

$$\hat{\Phi}_{p1}(k) = \hat{\Phi}_{p1}(k-1) + \frac{\eta\Delta u_p(k-1)}{\mu + |\Delta u_p(k-1)|^2}$$
$$(\Delta x_p(k+1) - \hat{\Phi}_{p1}(k-1)\Delta u_p(k-1))$$

where $\eta \in (0, 2]$ is the step-size factor, which enhances the algorithm's flexibility and generality, and $\mu$ is a positive constant.

Similarly, for System 2, the cost function is given in the following form:

$$J(\hat{\Phi}_{p2}(k)) = |\Delta\varepsilon_p(k+1) - \hat{\Phi}_{p2}(k)\Delta x_p(k-1)|^2 + \mu|\hat{\Phi}_{p2}(k) - \hat{\Phi}_{p2}(k-1)|^2$$

Taking the partial derivative of $\hat{\Phi}_{p2}(k)$ from the above equation yields 0, with

$$\hat{\Phi}_{p2}(k) = \hat{\Phi}_{p2}(k-1) + \frac{\eta\Delta x_p(k-1)}{\mu + |\Delta x_p(k-1)|^2}$$
$$(\Delta\varepsilon_p(k+1) - \hat{\Phi}_p(k-1)\Delta x_p(k-1))$$

Based on this, a control protocol is formulated for System 1 and System 2,

$$u_p(k) = u_p(k-1) + \frac{\rho\hat{\Phi}_{p1}(k)}{\lambda + \left|\hat{\Phi}_{p1}(k)\right|^2}$$
$$(\hat{x}_p(k) - x_p(k))$$

$$x_p(k) = x_p(k-1) + \frac{\rho\hat{\Phi}_{p2}(k)}{\lambda + \left|\hat{\Phi}_{p2}(k)\right|^2}\bar{\varepsilon}_p(k)$$

Based on the current position and current time velocity state, the virtual desired velocity state is designed as follows:

$$\begin{cases} \hat{x}_p(k) = \hat{x}_p(k-1) + \frac{\rho\hat{\Phi}_{p1}^2(k)}{\lambda + \hat{\Phi}_{p1}^2(k)}\tilde{x}_p(k-1) + g \\ g = -\tilde{x}_p(k-1) \end{cases}$$

Therefore, a complete control framework is as follows

$$\hat{\Phi}_{p1}(k) = \hat{\Phi}_{p1}(k-1) + \frac{\eta\Delta u_p(k-1)}{\mu + |\Delta u_p(k-1)|^2} \tag{12}$$
$$(\Delta x_p(k) - \hat{\Phi}_{p1}(k-1)\Delta u_p(k-1))$$

$$\hat{\Phi}_{p1}(k) = \hat{\Phi}_{p1}(1) \text{ if } \left|\hat{\Phi}_{p1}(k)\right| \leq \xi \text{ or } |\Delta u_p(k-1)| \leq \xi$$

$$u_p(k) = u_p(k-1) + \frac{\rho\hat{\Phi}_{p1}(k)}{\lambda + \left|\hat{\Phi}_{p1}(k)\right|^2}(\hat{x}_p(k) - x_p(k)) \tag{13}$$

$$\hat{\Phi}_{p2}(k) = \hat{\Phi}_{p2}(k-1) + \frac{\eta^* \Delta x_p(k-1)}{\mu^* + |\Delta x_p(k-1)|^2} (\Delta \bar{\varepsilon}_p(k)$$
$$- \hat{\Phi}_{p2}(k-1) \Delta x_p(k-1)) \tag{14}$$

$$\hat{\Phi}_{p2}(k) = \hat{\Phi}_{p2}(1) \text{ if } \left| \hat{\Phi}_{p2}(k) \right| \le \xi^* \text{ or } |\Delta x_p(k-1)| \le \xi^*$$

$$x_p(k) = x_p(k-1) + \frac{\rho \hat{\Phi}_{p2}(k)}{\lambda^* + \left| \hat{\Phi}_{p2}(k) \right|^2} \bar{\varepsilon}_p(k) \tag{15}$$

$$\begin{cases} \hat{x}_p(k) = \hat{x}_p(k-1) + \frac{\rho \hat{\Phi}_{p1}^2(k)}{\alpha + \hat{\Phi}_{p1}^2(k)} \tilde{x}_p(k-1) + g \\ g = -\tilde{x}_p(k-1) \end{cases} \tag{16}$$

The above MFAC consists of PPD estimation algorithm(12) (14), PPD reset algorithm, control algorithm(13) (15), and virtual estimation algorithm(16), which is designed with compensated distributed output $\Delta \bar{\varepsilon}_p(k)$.

### C. Convergence Analysis

In this section, the convergence analysis of the system (1) that has suffered stochastic cyber-attacks will be discussed, and the main contents are concluded in Theorem 2.

*Theorem 2:* For system (1) subjected to stochastic cyber-attacks, if assumptions 1-4 are satisfied and controller parameters are chosen as $\lambda > 0$, $0 < \eta < 2$, $\mu > 0$, $\rho \in (0,1]$, the consensus control target can be achieved by using control algorithms (12)-(16).

*Proof:* The proof will be introduced in four parts. First, it is shown that the approximation error of the PPD parameter is bounded. Second, the boundedness of the error in virtual tracking speed is demonstrated. Third, the bounded nature of the distributed output prediction error is proven. Fourth, the boundedness of the distributed output error is demonstrated. The aforementioned bounded properties are all based on the concept of mathematical expectation.

*part-1:* Let $\hat{\Phi}_{p1}(k) = \hat{\Phi}_{p1}(k) - \Phi_{p1}(k)$ be the approximation error of the PPD parameter, according to (12).

$$\tilde{\Phi}_{p1}(k) = \left( 1 - \frac{\eta \Delta u_p(k-1)^2}{\mu + |\Delta u_p(k-1)|^2} \right) \times \tilde{\Phi}_{p1}(k-1) + \Phi_{p1}(k-1) - \Phi_{p1}(k). \tag{17}$$

We can obtain from (20) that

$$|\tilde{\Phi}_{p1}(k)| \le \left| \left( 1 - \frac{\eta \Delta u_p(k-1)^2}{\mu + |\Delta u_p(k-1)|^2} \right) \right| |\tilde{\Phi}_{p1}(k-1)| + |\Phi_{p1}(k-1) - \Phi_{p1}](k)|. \tag{18}$$

Since $|\Delta u_p(k)| \le b$, by appropriately selecting $\eta$ and $\mu$ such that $0 < \eta \le 1$ and $\mu \ge 0$, there exists a constant $q_1$ ensuring that holds.

$$0 < \left( 1 - \frac{\eta \Delta u_p(k-1)^2}{\mu + |\Delta u_p(k-1)|^2} \right) \le q_1 < 1 \tag{19}$$

Since $|\Phi_{p1}(k)| \le a$, considering Assumption 4, we can obtain $|\Phi_{p1}(k-1) - \Phi_{p1}(k)| \le a$.

From (8) and (9), we have

$$|\tilde{\Phi}_{p1}(k)| \le q_1 |\tilde{\Phi}_{p1}(k-1)| + a$$
$$\le \cdots$$
$$\le q_1^k |\hat{\Phi}_{p1}(0)| + \frac{a(1-q_1^k)}{1-q_1} \tag{20}$$

which implies that $\tilde{\Phi}_{p1}(k)$ is bounded. Consequently, $\Phi_{p1}(k)$ is also bounded as $\tilde{\Phi}_{p1}(k)$ is bounded.

*part-II:* Let $\tilde{\Phi}_{p2}(k) = \hat{\Phi}_{p2}(k) - \Phi_{p2}(k)$ be the approximation error of PPD parameter, according to (14)

$$\tilde{\Phi}_{p2}(k) = \tilde{\Phi}_{p2}(k-1) - \Delta \Phi_{p2}(k) + \frac{\eta^* \Delta x_p(k-1)}{\mu^* + |\Delta x_p(k-1)|^2}$$
$$(\bar{\varepsilon}_p(k) - \bar{\varepsilon}_p(k-1) - \hat{\Phi}_{p2}(k-1) \Delta x_p(k-1))$$
$$= \tilde{\Phi}_{p2}(k-1) - \Delta \Phi_{p2}(k) + \frac{\eta^* \Delta x_p(k-1)}{\mu^* + |\Delta x_p(k-1)|^2}$$
$$(l(k)\varepsilon_p(k) - l(k)\hat{\varepsilon}_p(k))$$
$$= [1 - \frac{\eta^* \Delta x_p^2(k-1) l(k)}{\mu^* + |\Delta x_p(k-1)|^2}] \tilde{\Phi}_{p2}(k-1) - \Delta \Phi_{p2}(k) \tag{21}$$

By taking the absolute value and the mathematical expectation on both sides of the above formula, then

$$\mathbb{E}\{|\tilde{\Phi}_{p2}(k)|\} \le |1 - \frac{\eta^* \Delta x_p^2(k-1) l(k)}{\mu^* + \Delta x_p^2(k-1)}| |\mathbb{E}\{|\tilde{\Phi}_{p2}(k-1)|\}$$
$$+ |\Delta \Phi_{p2}(k)| \tag{22}$$

Notice that for $0 < \eta^* < 2$, $\mu^* > 0$, $0 < l(k) < 1$. Therefore, there is a positive constant d that makes $0 < |1 - \frac{\eta^* \Delta x_p^2(k-1) l(k)}{\mu^* + \Delta u^2(k-1)}| = d < 1$. Also because of $|\Phi_{p2}(k)| \le \mathfrak{b}_2$, so $|\triangle \Phi_{p2}(k)| \le 2\mathfrak{b}_2$. (22) can be expressed as,

$$\mathbb{E}\{|\tilde{\Phi}_{p2}(k)|\} \le d \mathbb{E}\{|\tilde{\Phi}_{p2}(k-1)|\} + 2\mathfrak{b}_2$$
$$\le \cdots$$
$$\le d^{k-1} \mathbb{E}\{|\tilde{\Phi}_{p2}(1)|\} + \frac{2\mathfrak{b}_2}{1-d} \tag{23}$$

From the above inequality (26), we understand that $\tilde{\Phi}_{p2}(k)$ is uniformly bounded. Therefore, based on $\Phi_{p2}(k)$ being bounded, $\hat{\Phi}_{p2}(k)$ is also bounded.

*part-III:* Define the virtual desired velocity tracking error $\tilde{x}_p(k) = \hat{x}_p(k) - x_p(k)$. Based on equations (15) and (16), it can be expressed as,

$$\tilde{x}_p(k+1) = \hat{x}_p(k+1) - \hat{x}_p(k) + \hat{x}_p(k+1) - x_p(k+1)$$
$$= \Delta \hat{x}_p(k+1) + (1 - \frac{\rho \hat{\Phi}_{p1}(k) \Phi_{p1}(k)}{\alpha + \hat{\Phi}_{p1}^2(k)}) \tilde{x}_p(k) \tag{24}$$

Let a Lyapunov function be defined:$v(k) = \tilde{x}_p^2(k)$

$$\Delta v(k+1) = (\tilde{x}_p(k+1) - \tilde{x}_p(k))(\tilde{x}_p(k+1) + \tilde{x}_p(k))$$

$$= 2\tilde{x}_p(k)(\Delta\tilde{x}_p(k+1) - \frac{\rho\hat{\Phi}_{p1}(k)\Phi_{p1}(k)}{\alpha + \hat{\Phi}_{p1}^2(k)}\tilde{x}_p(k))$$

$$(\Delta\tilde{x}_p(k+1) - \frac{\rho\hat{\Phi}_{p1}(k)\Phi_{p1}(k)}{\alpha + \hat{\Phi}_{p1}^2(k)}\tilde{x}_p(k))^2 \quad (25)$$

when $\lim_{k\to\infty} a = 0$, so $\lim_{a\to 0 k\to\infty} \tilde{\Phi}_{p1}(k) = 0$

$$\Delta v(k+1) = 2\tilde{x}_p(k)(g + \frac{\rho\hat{\Phi}_{p1}(k)\tilde{\Phi}_{p1}(k)}{\alpha + \hat{\Phi}_{p1}^2(k)}\tilde{x}_p(k))$$

$$(g + \frac{\rho\hat{\Phi}_{p1}(k)\tilde{\Phi}_{p1}(k)}{\alpha + \hat{\Phi}_{p1}^2(k)}\tilde{x}_p(k))^2$$

$$= g^2 + 2\tilde{x}_p(k)g$$

$$= -\tilde{x}_p^2(k) \quad (26)$$

It is evident that, under the control algorithm proposed in this paper, $\Delta v(k) < 0$ is consistently satisfied. The disparity between the actual velocity and the virtual desired velocity converges asymptotically and steadily, thus completing the proof.

*part-IV:* The distributed output prediction error is defined as $\bar{\varepsilon}_p(k) = \hat{\varepsilon}_p(k) - \varepsilon_p(k)$. Based on equations (11), it can be expressed as follows:

$$\tilde{\varepsilon}_p(k) = \hat{\varepsilon}_p(k) - \varepsilon_p(k) \quad (27)$$

$$= \bar{\varepsilon}_p(k-1) + \hat{\Phi}_{p2}(k-1)\triangle x_p(k-1)$$

$$- \varepsilon_p(k-1) - \Phi_{p2}(k-1)\Delta x_p(k-1)$$

$$= (1 - l(k))\tilde{\varepsilon}_p(k-1) + \hat{\Phi}_{p2}(k-1)$$

$$\triangle x_p(k-1) \quad (28)$$

Take the absolute value and expectation of the formula for (27), and scale them to obtain the result , then

$$\mathbb{E}\{|\tilde{\varepsilon}_p(k)|\} \le (1 - l(k))\mathbb{E}\{|\tilde{\varepsilon}_p(k-1)|\}$$

$$+ |\tilde{\Phi}_{p2}(k-1)\triangle x_p(k-1)| \quad (29)$$

Due to the boundedness of $\hat{\Phi}_{p1}(k)$ proven in the part-I, and from our findings in the part-II that $x_p(k-1)$ is also bounded, we can conclude that $\hat{\Phi}_{p1}(k)x_p(k-1)$ is bounded.

Let $\hat{\Phi}_{p1}(k)x_p(k-1) < q$, where $q$ is a positive constant. Given that $l \in (0,1)$, the inequality can then be re-expressed as:

$$\mathbb{E}\{|\tilde{\varepsilon}_p(k)|\} \le (1 - l)\mathbb{E}\{|\tilde{\varepsilon}_p(k-1)|\} + q \quad (30)$$

$$\le \cdots$$

$$\le (1 - l)^{k-1}\mathbb{E}\{|\tilde{\varepsilon}_p(1)|\} + \frac{q}{1 - l} \quad (31)$$

which means that the $\tilde{\varepsilon}_p(k)$ is uniformly bounded.

*part-V:* Due to (14) and $e_{\varepsilon_p}(k+1) = \varepsilon_p^* - \varepsilon_p(k-1)$, $e_{\varepsilon_p}(k+1)$ can be expressed as,

$$e_{\varepsilon_p}(k+1) = \varepsilon_p^* - \varepsilon_p(k-1)$$

$$= e_{\varepsilon_p}(k) - \Delta\varepsilon_p(k-1)$$

$$= (1 - \frac{\rho^*\hat{\Phi}_{p2}(k)\Phi_{p2}(k)}{\lambda^* + \hat{\Phi}_{p2}^2(k)})e_{\varepsilon_p}(k)$$

$$+ \frac{\rho^*\hat{\Phi}_{p2}(k)\Phi_{p2}(k)}{\lambda^* + \hat{\Phi}_{p2}^2(k)}\hat{\varepsilon}_p(k) \quad (32)$$

let $\frac{\rho^*\hat{\Phi}_{p2}(k)\Phi_{p2}(k)}{\lambda^* + \hat{\Phi}_{p2}^2(k)} = m(k)$

Using the same processing method as in part-I, equation (35) can then be rewritten as follows:

$$|e_{\varepsilon_p}(k+1)| \le |1 - m(k)|\,|e_{\varepsilon_p}(k)|$$

$$|m(k)|\,\left|l(k)\tilde{\varepsilon}_p(k) + \tilde{\Phi}_{p2}(k-1)\Delta x_p(k-1)\right| \quad (33)$$

According to the part I and part II proofs, $\tilde{\varepsilon}_p(k)$ and $\tilde{\Phi}_{p2}(k-1)$ are bounded, so let $(1-D)l\mathbb{E}\{|\tilde{\varepsilon}_p(k)|\} + (1-D)\mathbb{E}\{|\tilde{\Phi}_{p2}(k-1)\Delta x_p(k-1)|\} < d$ with $d > 0$ being a constant. Then, the (29) can be expressed as,

$$\mathbb{E}\{|e_{\varepsilon_p}(k+1)|\} \le D\mathbb{E}\{|e_{\varepsilon_p}(k)|\} + (1-D)l\mathbb{E}\{|\tilde{\varepsilon}_p(k)|\}$$

$$(1-D)\mathbb{E}\{|\tilde{\Phi}_{p2}(k-1)\Delta x_p(k-1)|\}$$

$$\le D\mathbb{E}\{|e_{\varepsilon_p}(k)|\} + d$$

$$\le D^k\mathbb{E}\{|e_{\varepsilon_p}(1)|\} + \frac{d}{1-D} \quad (34)$$

which illustrate $\varepsilon_p^*$ is bounded. Due to $e_{\varepsilon_p}(k+1)$ is bounded, $\varepsilon_p(k-1)$ is also bounded.

This is the end of the proof,this crucial result ensures that the proposed control algorithm effectively maintains system stability and achieves its intended control objectives under the specified conditions. The meticulous application of Lyapunov functions and the detailed step-by-step analysis provide a solid theoretical foundation for our approach. This comprehensive proof not only validates the robustness and reliability of the proposed control strategy but also highlights its practical applicability in real-world scenarios where multi-agent systems are subject to mixed network attacks. Consequently, we can confidently assert that the control strategy developed in this paper is both sound and effective, offering significant potential for enhancing the performance and resilience of multi-agent systems in dynamic and potentially hostile environments.

## IV. SIMULATION

A numerical case is presented in this section to validate the effectiveness of the proposed approach.

*Example 1:* Consider a MASs comprising one leader and four followers, with its communication topology graph shown in Fig. 1.

The dynamic model of each agent is expressed as follows:

$$\begin{cases} Agent1 : y_1(k+1) = \dfrac{y_1(k)x_1(k)}{1+y_1^2(k)} + \dfrac{x_1(k)u_1(k)}{1+x_1^2(k)} + u_1(k) \\ Agent2 : y_2(k+1) = \dfrac{y_2(k)x_2(k)}{1+y_2^2(k)} + \dfrac{x_2(k)u_2(k)}{1+x_2^3(k)} + 0.5u_2(k) \\ Agent3 : y_3(k+1) = \dfrac{y_3(k)x_3(k)}{1+y_3^2(k)} + \dfrac{x_3(k)u_3(k)}{1+x_3^2(k)} + 0.9u_3(k) \\ Agent4 : y_4(k+1) = \dfrac{y_4(k)x_4(k)}{1+y_4^2(k)} + \dfrac{x_4(k)u_4(k)}{1+x_4^5(k)} + 0.8u_4(k) \end{cases}$$

In addition, the Laplacian matrix can be derived from the communication topology graph shown in Fig. 2, where

$$\mathcal{L} = \begin{bmatrix} 1 & 0 & 0 & -1 \\ 0 & 1 & -1 & 0 \\ 0 & -1 & 2 & -1 \\ 0 & -1 & -1 & 2 \end{bmatrix}$$

The trajectory of the leader is given by the following formula:

$$y_d(k) = \begin{cases} 2, & 0 < k \leq 200 \\ 0.5, & 200 < k \leq 400 \end{cases}$$

Here are the initial values and controller parameters for the agents. $\bar{l}_1 = 0.3$, $\bar{l}_2 = 0.4$, $\bar{l}_3 = 0.35$, $\bar{l}_4 = 0.45$ and The gain parameter for FDI attacks $\pi_p(k)$ varies randomly in the range of $[0, 5]$. The initial values and controller parameters are set as follows $y_p(0) = [1; 1; 1; 1]$, $u_p(0) = [1; 1; 1; 1]$, $\hat{\Phi}_{p1}(k) = [1.05; 1.1; 1.2; 1.03]$, $\hat{\Phi}_{p2}(k) = [1.2; 1.1; 1.02; 1.3]$, $\xi = 10^{-5}$, $\rho = 0.3$, $\eta = 1.5$, $\lambda = 4$, $\mu = 0.5$, $p = 1, 2, 3, 4$. $\rho^* = 0.3$, $\eta^* = 1.5$, $\lambda^* = 4$, $\mu^* = 0.5$, $p = 1, 2, 3, 4$. Simulation results are displayed in Fig. 3 to 6. Fig. 3 shows the tracking performance under a time-invariant signal in Example 1, comparing the proposed algorithm, a reference algorithm, and the performance without compensation. Fig. 5 illustrates the tracking performance under a time-varying signal in Example 2, using the proposed algorithm, a comparative algorithm, and an uncompensated scenario. Fig. 4 . 6 . 7 and 8 present the error trajectories for the systems under the proposed and comparative algorithms. This clear comparison demonstrates that the proposed method yields good control effects in simulations.

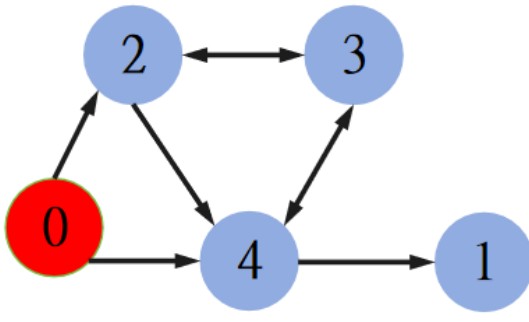

Fig. 2. Topology graph of Examples 1 and 2.

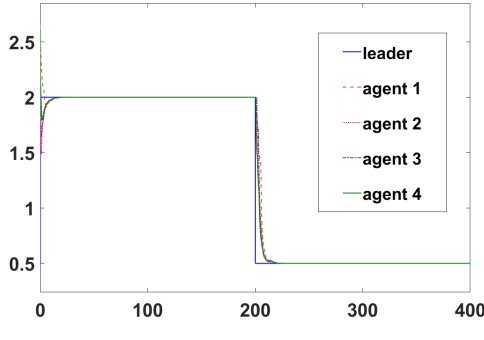

(a)

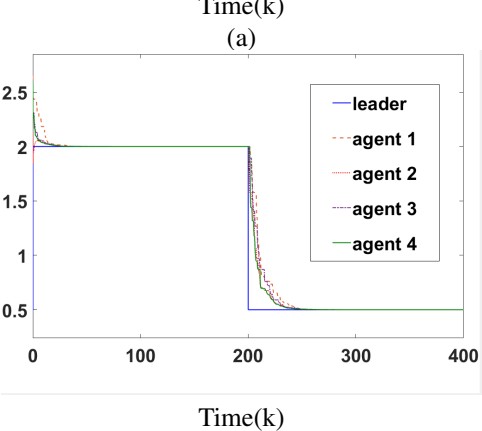

(b)

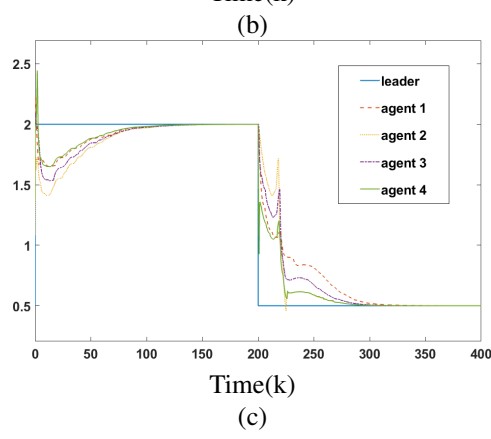

(c)

Fig. 3. Performance of multi-agent systems tracking under mixed attacks. (a) Proposed MFAC algorithm. (b) Control methodology described in reference [24]. (c) Control strategy without any compensation.

*Remark 4:* In the simulation, the expected trajectory of the virtual leader is defined and set as node 0. In this scenario, only agents 2 and 4 can directly receive information from the leader, while agents 1 and 3 can only indirectly receive information from the leader through agents 2 and 4.

The aforementioned simulation results indicate that, under time-invariant signals, the proposed MFAC algorithm exhibits superior control performance compared to the control method described in reference [24]. Specifically, the proposed algorithm achieves smaller errors and higher stability. Furthermore, the compensation method introduced in this paper demonstrates a significant improvement over the scenario without com-

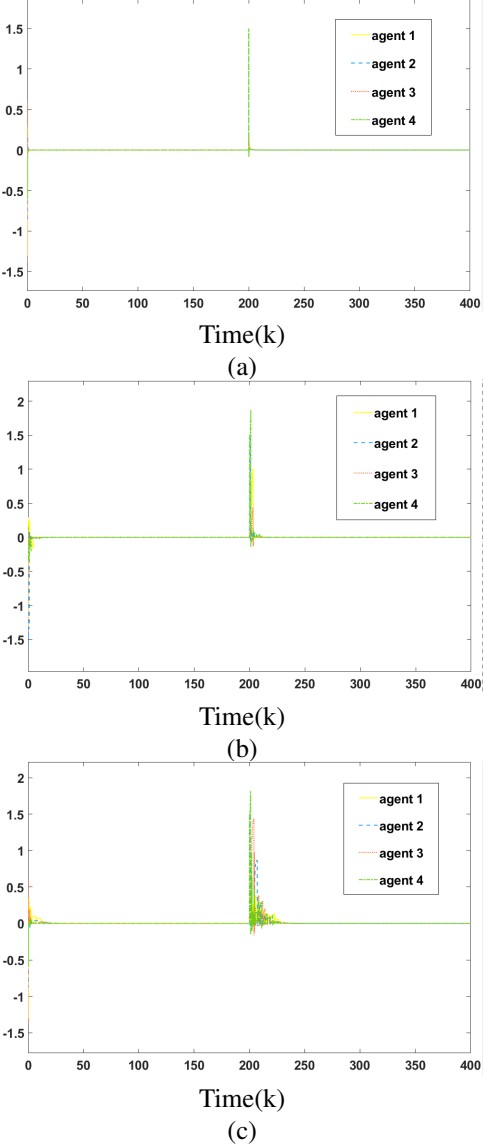

Fig. 4. The tracking error of multi-agent systems under mixed attacks. (a) Proposed MFAC algorithm. (b) Control methodology described in reference [24]. (c) Control strategy without any compensation.

pensation. The control effect with compensation is markedly better than that without compensation. In summary, the MFAC algorithm with compensation proposed in this paper shows excellent control performance under time-invariant signals, making it a robust and effective solution for managing system stability and accuracy.

*Example 2:* A multi-agent system with a time-varying signal is analyzed, with its communication topology graph depicted in Fig. 1.

The trajectory of the leader is given by the following formula:

$$y_d(k) = 0.6 + 0.2(\sin(\frac{2\pi k}{50}) + \sin(\frac{2\pi k}{100}) + \sin(\frac{2\pi k}{150}))$$

Here are the initial values and controller parameters for the

agents. $\bar{l}_1 = 0.3$, $\bar{l}_2 = 0.4$, $\bar{l}_3 = 0.35$, $\bar{l}_4 = 0.45$ and The gain parameter for FDI attacks $\pi_p(k)$ varies randomly in the range of $[0, 5]$. The initial values and controller parameters are set as follows $y_p(0) = [0; 0; 0; 0]$, $u_p(0) = [1; 1; 1; 1]$, $\hat{\Phi}_{p1}(0) = [1.05; 1.1; 1.02; 1.3]$, $\hat{\Phi}_{p2}(0) = [1.2; 1.1; 1.02; 1.3]$, $\xi = 10^{-5}$, $\rho = 0.3$, $\eta = 1.5$, $\lambda = 4$, $\mu = 0.5$, $p = 1, 2, 3, 4$. $\rho^* = 0.3$, $\eta^* = 1.5$, $\lambda^* = 4$, $\mu^* = 0.5$, $p = 1, 2, 3, 4$. with simulation results shown in Fig. 4. Simulation results demonstrate that, whether under time-varying or time-invariant signals, the proposed algorithm exhibits smaller errors, faster convergence, and better tracking performance compared to the algorithm presented in [24]. Through clear comparisons across various aspects, the proposed method shows good control effects in simulations. The simulation results illustrated above

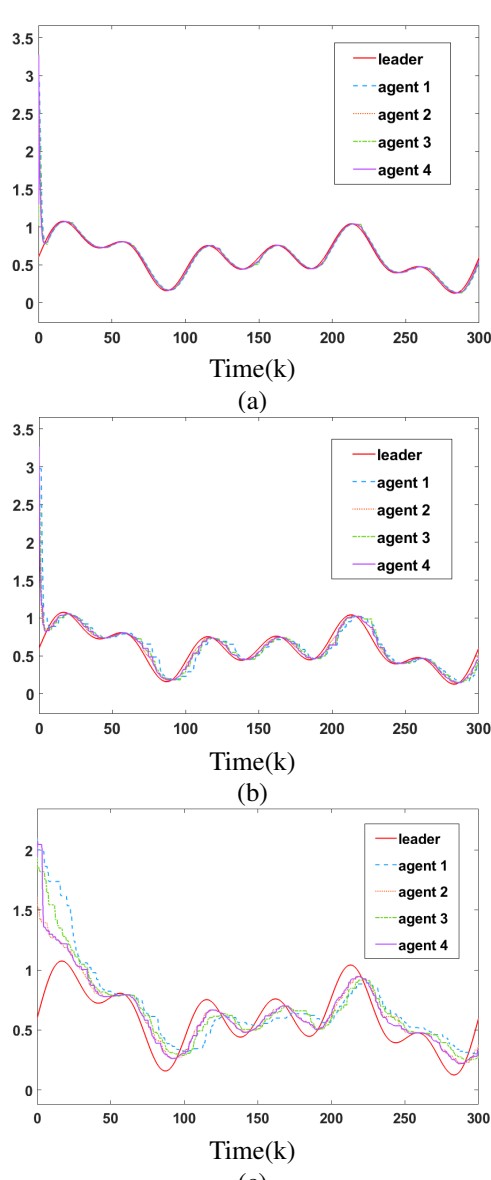

Fig. 5. Tracking performance of multi-agent systems under mixed attacks. (a) The proposed MFAC algorithm. (b) The control algorithm in reference [24]. (c) The control algorithm without compensation.

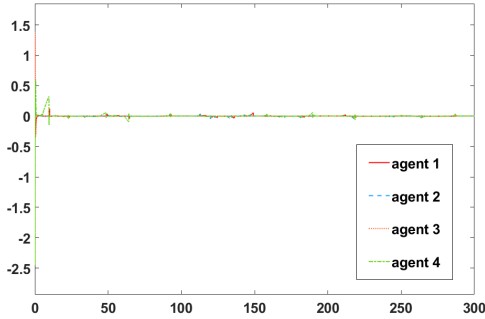

Fig. 6. Error trajectories $e_{\varepsilon_p}(k)(p=1,2,3,4)$ of the proposed algorithm.

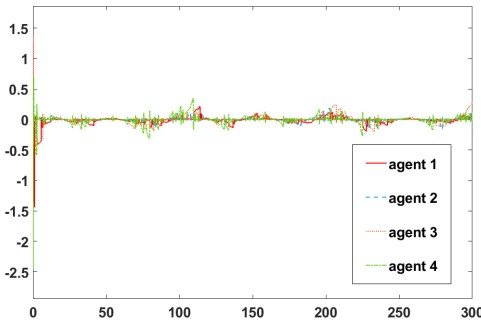

Fig. 7. Error trajectories $e_{\varepsilon_p}(k)(p=1,2,3,4)$ of the algorithm proposed in reference [24].

indicate that, under time-varying signals, the proposed MFAC algorithm demonstrates markedly better control performance compared to the control strategy outlined in reference [24]. This improvement is reflected in significantly reduced error margins and enhanced system stability. Moreover, the introduction of the compensation method proposed in this study shows a clear and substantial improvement over the scenario without compensation. Specifically, the control effect achieved with compensation is considerably superior to that without compensation. Thus, it can be concluded that the MFAC algorithm, when coupled with the proposed compensation technique, delivers outstanding control performance under time-varying signal conditions. This highlights the algorithm's robustness and efficacy in managing dynamic systems with fluctuating signals, ensuring both precision and stability in

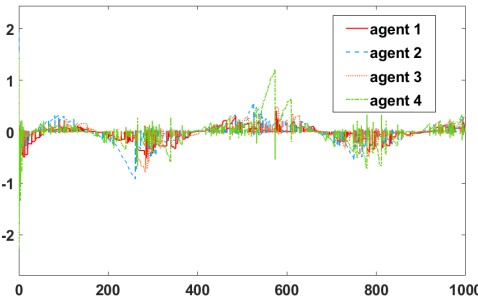

Fig. 8. Error trajectories $e_{\varepsilon_p}(k)(p=1,2,3,4)$ of the algorithm without compensation.

control outcomes.

## V. CONCLUSION

This paper investigates the model-free adaptive control problem of second-order nonlinear multi-agent systems under mixed attacks. To facilitate subsequent research, the distributed output error of each agent in each subsystem is defined based on backstepping and model-free techniques, then transformed into a linear dynamic model using the dynamic linearization method. This transformation simplifies the control design process and makes it more manageable. Moreover, the design of mixed network attacks, including denial of service (DoS), false data injection (FDI), and deception attacks, is elaborated, and an attack compensation mechanism is proposed to mitigate their impact on the system. This mechanism ensures the systems robustness against various types of network attacks. The convergence of the proposed control strategy is then demonstrated based on Lyapunov functions, providing a solid theoretical foundation for the approach. Finally, numerical simulations demonstrate that this scheme exhibits good control performance for the system, maintaining stability and effectiveness even under challenging conditions.

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
