# OpenReview forum: "Model free adaptive control of nonlinear second-order multi-agent systems based on backstepping under mixed attacks"
_IEEE.org/ICIST/2024/Conference — IEEE ICIST 2024 Conference Submission_

### Official Review · Reviewer_JqhF · 2024-08-27
**Comments for Improvement**

**Rating:** 10
**Confidence:** 4

**Review:**

The paper presents a sophisticated approach to addressing the trajectory tracking control problem in multi-agent systems (MASs) under mixed network attacks using a model-free adaptive control (MFAC) method. Here are some comments for improving the equality of the paper:

1.	The lines in Fig. 6 are not clear.

2.	Potential and future research directions could be added to the conclusion.

---

### Official Review · Reviewer_6d8X · 2024-08-30
**The research is innovative**

**Rating:** 9
**Confidence:** 5

**Review:**

1. While the paper presents a novel approach using model-free adaptive control (MFAC) and backstepping methods, it would benefit from a more detailed explanation of the proposed methodology. Specifically, additional clarity on how the virtual desired velocities are constructed and how the two interconnected subsystems are designed could enhance reader understanding.
2. The paper validates the proposed method through two simulation examples. It would be valuable to see a broader range of simulation scenarios that examine different conditions, such as varying degrees of attack intensity and different network configurations. This would help demonstrate the robustness of the proposed approach under diverse situations.

---

### Official Review · Reviewer_H1g9 · 2024-09-01
**This manuscript addresses the issue of model-free adaptive control for second-order nonlinear multi-agent systems subject to mixed attacks.**

**Rating:** 5
**Confidence:** 5

**Review:**

1.Ensuring convergence of a model-free adaptive control algorithm to the desired control strategy and maintaining system stability is a major challenge. Without a model, how can one guarantee that the control actions will meet the desired outcomes in all scenarios?
2.There are three main types of model-free adaptive control: compact-form dynamic linearization, partial-form dynamic linearization, and full-form dynamic linearization. Each method has its differences. Which one is adopted in this manuscript, why was it chosen, and what are its advantages and disadvantages?
3.The curves in Figures 3 to 8 are not clear, and it is suggested to redraw the graphs.
4.The format of the references is incorrect; please review and make the necessary corrections.
5.Errors are present in formulas (27) and (28); a thorough check of the manuscript is recommended.

---

### Decision · Program_Chairs · 2024-09-06

Accept (Oral)